# Effect of Flavorization on Virgin Olive Oil Oxidation and Volatile Profile

**DOI:** 10.3390/antiox12020242

**Published:** 2023-01-21

**Authors:** Enrique Jacobo Díaz-Montaña, Ramón Aparicio-Ruiz, María T. Morales

**Affiliations:** Department of Analytical Chemistry, Faculty of Pharmacy, University of Seville, Calle Profesor García González, 41012 Seville, Spain

**Keywords:** volatile compounds, virgin olive oil, flavored olive oil, rosemary, basil, oxidation

## Abstract

The volatile compounds of virgin olive oil (VOO) have an important role from a sensory point of view as they are responsible for the aroma of the oil. Once the oil is obtained, auto-oxidation is the main process contributing to its deterioration, modifying the volatiles profile and aroma. The addition of aromatic herbs to VOO is a traditional technique to change the flavor and to preserve the oil. The aim of this study was to evaluate the effect on the volatile profile and sensory properties of flavoring VOO with rosemary and basil herbs and its impact on the evolution of the oxidative process during a six-month shelf-life study at 15.7 ± 3.6 °C and exposed to 500 ± 100 lx of light for 12 h each day. The determination of quality parameters, volatiles concentrations and VOO sensory properties and their comparison with the flavored VOO samples showed that the addition of basil or rosemary herbs, in addition to retarding the oxidation of the oil, allowed the discrimination of the flavored samples due to the migration of compounds from herbs to the oil. The aroma of basil olive oil (BOO) samples was mainly due to β-pinene, ocimene and 1,8-cineol compounds while for rosemary olive oil (ROO) samples, their aroma was mainly due to the concentrations of camphene, β-myrcene, α-terpinolene, limonene and 1,8-cineol. From the antioxidant standpoint, the effect of the herbs was more noticeable from the third month onwards.

## 1. Introduction

Olive oil is the main fat and one of the dietary staples in countries of the Mediterranean basin. In recent years, the consumption of olive oil has increased worldwide due to its well-known health benefits which are mainly due to the presence of antioxidant compounds, among others. Virgin olive oil (VOO) is obtained from the fruit of the olive tree (*Olea europaea* L.) when it is fresh, sound and has its optimal maturity [1]. VOO, and extra virgin olive oil (EVOO) in particular, differs from other edible oils in its nutritional, health and organoleptic properties [1,2,3]; it is one of the most-studied antioxidant food sources [4]. The health properties of VOO are attributed to a high content of oleic acid (monounsaturated fatty acid) and the presence of a series of minor components such as triterpenic alcohols, phenols, tocopherols, etc. [1,5]. In recent years, more health properties of olive oil have been reported, including oxidative stress prevention, anti-inflammatory properties, lowering the risk of hypertension and antitumoral properties, among many others. These properties have allowed olive oil to stand out from other fats and edible oils [2].

VOO is also appreciated for its sensory characteristics; it has a pleasant aroma and delicious taste, which contribute to appetite, satiety and consumer preference [6]. Olive oil’s flavor is due to the presence of phenolic compounds which are responsible for its bitterness and pungency, and the volatile compounds, which are responsible for the aroma [4,7]. The biogenesis of VOO volatiles is mainly due to the oxidation of unsaturated fatty acids by the endogenous enzyme lipoxygenase, which generates volatile compounds using linoleic and linolenic acids as precursors [8]. C6 aldehydes and alcohols, and their corresponding esters, are mainly responsible for positive sensory perception, whereas chemical oxidation, hydrolysis and exogenous enzymes (usually from microorganisms) produce the volatiles associated with sensory defects [9]. The quality deterioration of oil is linked to food safety since oxidation products are formed when oil ages. The oxidative degradation of olive oil causes an increase in free radicals and the formation of off-flavors [10], producing the appearance of a rancid sensory defect [11] and a decrease in the antioxidant content [12]. The oxidation process is highly influenced by the processing and storage conditions of the oil, which affects not only its sensory and healthy qualities [13], but also its economic value and consumer acceptability [6].

The addition of aromatic herbs, bulbs or fruits—such as oregano, basil, rosemary, thyme, garlic and lemon—to olive oil results in aromatized or flavored olive oils (FOO) [14,15,16,17,18,19,20], thereby modifying their shelf life [12], and perhaps, masking undesirable sensory perceptions. The flavoring process of VOOs is varied and can include the infusion of spices into the oil, ultrasound-assisted maceration and joint malaxation of olive paste and herbs, among others [14,18,19,20,21]. Some studies have characterized the flavoring process through the detection of new sensory characteristics. Additionally, olive oil flavoring has also been studied because it enhances its health properties including antimicrobial, anticancer or antioxidant properties [22,23,24]. Thus, the VOO flavoring not only modifies the initial VOO sensory characteristics and health properties, but also can change its shelf life and quality parameters as consequence of the migration/transference of antioxidants and/or pro-oxidant compounds from the herbs to the oils [25,26].

Research is usually focused on characterizing flavored VOOs and only a few studies have been conducted on how these FOOs evolve over time [25,27,28,29]. Hence, the aims of this work were twofold: (i) to study and compare the changes produced on the profiles of volatile compounds in non-flavored olive oils and virgin oils flavored by the contact method with rosemary (*Rosmarinus officinalis* L.) and basil (*Ocimum basilicum* L.) during six months of storage at 15.7 ± 3.6 °C and exposed to 500 ± 100 lx of light for 12 h each day; and, (ii) to characterize the new volatiles appearing in flavored olive oils (FOO) and to study their quantitative changes and influence on the shelf life.

## 2. Materials and Methods

### 2.1. Chemicals

The reagents employed were of pure analytical grade. Acetic acid (99%), ethanol (96%) β-caryophyllene (80%), β-myrcene (90%), camphene (95%), α-pinene (98%), borneol (97%), 2-methylbutanal (95%), naphthalene (98%), p-cymene (99%), α-linalool (97%), β-farnesene (90%), β-pinene (99%), limonene (97%), 1,8-cineole (99%), octadecane (99%), γ-terpinene (97%), β-farnesene (90%), naphthalene (99%) and isobutyl acetate (<99.9%), used as internal standards, were purchased from Sigma-Aldrich (St. Louis, MO, USA).

Ethyl acetate (99.5%), butanoic acid (99%), propanoic acid (99.5%), pentanoic acid (99%), butan-1-ol (99%), butan-2-ol (99.5%), heptan-2-ol (99%), heptan-2-one (98%), (E)-2-heptenal (98%), hexanal (98%), hexan-1-ol (98%), (E)-2-hexenal (97%), 2-methyl-butan-1-ol (99.5%), 3-methyl-butan-1-ol (99.5%), 4-methyl-pentan-2-one (99%), nonanal (95%), octanal (98%), pentanal (98%), octane (99%), 1-penten-3-one (98%) and ethyl propanoate (99%) were purchased from Fluka (Zwijndrecht, The Netherlands).

Ethyl butanoate (98%), heptanal (95%), (E)-3-hexen-1-ol (98%), ethyl 2-methylbutanoate (99%), 6-methyl-5-hepten-2-one (99%), octan-3-one (97%), 1-octen-3-one (50%) and pentan-3-one (99.5%) were purchased from Panreac (Barcelona, Spain).

### 2.2. Samples and Storing Conditions

An EVOO (var. Manzanilla) from the 2020/21 harvest was purchased from a local market (Sevilla, Andalusia, Spain). The oil was filtered and its regulated quality parameters were determined [30]. Next, the sample (VOO-0) was divided into 18 aliquots stored in 1 L glass containers, 6 of them contained only EVOO (VOO), 6 contained EVOO dressed with dried rosemary (ROO, Rosemary Olive Oil) leaves at 5% *w*/*w* and another set of 6 samples contained EVOO dressed with dried basil (BOO, Basil Olive Oil) leaves at 5% *w*/*w* during the whole storage conditions, respectively. The aromatic herbs, collected in September of 2021 from the countryside of Seville (Andalusia, Spain) (37.320471, −5.979523), were treated to separate the leaves and discarded wooden parts, drying only the leaves at 60 °C for 10 h [31].

The 18 samples were stored for six months at 15.7 ± 3.6 °C and exposed to 500 ± 100 lx of light for 12 h each day. Once per month the volatile compounds, physical–chemical parameters and sensory characteristics were determined in triplicate by sampling one bottle from each group (VOO, ROO, BOO).

### 2.3. Quality Parameters

Free acidity (FA), peroxide value (PV) and specific extinction values (K232, K270 and ∆K) were determined according to The International Olive Council (IOC) trade standards [32].

### 2.4. Sensory Analysis

VOO-0 was evaluated according to [33] to assign the initial category of the purchased olive oil [30]. Evaluation was based on the determination of the median of the fruity attributes and the detection, or not, of sensory defects [33]. In addition, a sensory evaluation of all the samples was carried out by Quantitative Descriptive Analysis (QDA) to assess possible sensory changes occurring during storage [34]. For this purpose, a protocol was developed for the study including an evaluation sheet (Appendix A), which was created considering the different sensory attributes, and an unstructured 5 cm scale was used to evaluate the intensity of the attributes. The sensory evaluation was carried out by 8 assessors, all of whom were trained panelists, to check the consistency of the results.

### 2.5. Volatile Compounds Determination

Solid Phase Micro Extraction-Gas Chromatography-Mass spectrometry (SPME-GC-MS, Varian 3900 GC, Bruker, Palo Alto, CA, USA; MSD 5975, Agilent Tech, Santa Clara, CA, USA) and Dynamic Head Space-Gas Chromatography with Flame Ionization Detector (DHS-GC-FID, Varian 3900 GC, Bruker, Palo Alto, CA, USA) were employed to determine the volatile compounds.

#### 2.5.1. SPME-GC-MS

For SPME-GC-MS, 2 g of oil or herb samples was placed in a 30 mL glass vial tightly capped with silicone/polytetrafluoroethylene (PTFE) septum. To reach oil–air equilibrium, the samples were left for 10 min at 40 °C, after which a 1 cm SPME fiber with a 50/30 μm film thickness with divinylbenzene/carboxen/polydimethylsiloxane (DVB/CAR/PDMS) as a stationary phase, purchased from Supelco (Bellefonte, PA, USA), was exposed to the headspace for 40 min. The fiber was previously conditioned following the supplier instructions.

Desorption of volatiles was carried out in the hot injection port by heating the fiber for 5 min at 280 °C. The volatile compounds were injected into a TR-WAX capillary column (60 m × 0.25 mm i.d., 0.25 μm; Teknokroma, Barcelona, Spain), in splitless mode, of a Varian 3900 gas chromatograph (Bruker, Palo Alto, CA, USA) coupled to a mass spectrometer, using hydrogen at 0.9 mL/min as a carrier gas. The oven was kept in isothermal mode for 10 min at 35 °C and then increased by 3 °C/min until 200 °C. The GC–MS interface was heated to 280 °C with the actual temperature reaching 180 °C in MS source and 150 °C in MS-quadrupole [35]. The energy used for electron impact was set at 70 eV, and data were collected in the range of 40–300 atomic mass units (amu). The signal was recorded, and the integration was carried out with Enhanced Chemstation MSD ChemStation E.02.02.1431 (Agilent Tech, Santa Clara, CA, USA). For the volatile identification the library, Wiley 7 was used (John Wiley & Sons Limited, Hoboken, NJ, USA), and later confirmed by pure standards, when available.

#### 2.5.2. DHS-GC-FID

The dynamic headspace employed 1.5 g of samples. Once the samples were placed into a 20 mL glass vial and tightly capped with silicone/polytetrafluoroethylene (PTFE) septum, they were pre-heated for 18 min at 40 °C and mixed for 15 min using a HT3 Dynamic Headspace System (Teledyne Tekmar, Mason, OH, USA). After mixing, helium employed as a portable gas, using a flow rate of 5 mL/min, moved the volatile compounds to the adsorbent trap (Tenax TA^TM^, Supelco, Bellefonte, PA, USA).

The volatiles were thermally desorbed from the trap (5 min at 260 °C) and then injected with a split mode 7:1. As in the case of SPME, the injection was carried out in a TR-WAX capillary column (60 m × 0.25 mm i.d., 0.25 μm; Teknokroma, Barcelona, Spain) of a Varian 3900 gas chromatograph (Bruker, Palo Alto, CA, USA) with a flame ionization detector (FID), using hydrogen with a flow rate of 1.5 mL/min. The oven conditions were 10 min at 35 °C and then programmed to rise to 200 °C at 3 °C/min. The temperature of the flame ionization detector (FID) was set at 280 °C, and the signal was recorded with Star Chromatography Workstation, System Control version 6 (Bruker, Palo Alto, CA, USA).

Quantification was carried out using the internal standard methodological calibration procedure. The internal standard used was isobutyl acetate.

### 2.6. Statistical Analysis

The whole dataset was imported into Excel from the instrument and the Statistica 8 package (StatSoft, Tulsa, OK, USA) was used to carry out the data treatment and the statistical analyses. A t-test was used, which allowed the selection of compounds that showed significant differences (*p* < 0.05) in their relative concentration among the groups. Cluster analysis of the VOO samples, using the information of the total set of volatiles, was performed by a complete standardization of the data and was implemented with the City-Block (Manhattan) distance measure and the Ward’s method linkage amalgamation rule. Multidimensional Scaling (MDS)—an unsupervised statistical procedure—was implemented to explore the level of similarity of individual samples of the flavored and unflavored oils during the six-month storage experiment. The MDS algorithm is based on the results of the Cluster Analysis [36].

## 3. Results and Discussion

Rosemary and basil are usually used for VOO flavoring and to take advantage of its antioxidant properties, which could slow down the auto-oxidative process of VOO [37]. The volatile compounds of VOO vary according to different factors, such as fruit ripeness, pedoclimatic conditions, cultivation, time and conditions of storage, among others [38]. Particularly, virgin olive oil var. Manzanilla is characterized as having a different amount of alcohol dehydrogenase than other varieties, which leads to an accumulation of aldehydes. This variety is also characterized by having a low taste intensity related to bitterness and pungency, as well as a low aroma intensity of the fruity attribute, which make flavoring this oil a good option [39,40].

### 3.1. Quality Parameters

The initial sample (VOO-0), filtered and without storing, was classified as extra virgin olive oil once its physico-chemical and organoleptic-regulated parameters were determined [30]. VOO samples did not overstep the limits of any of the physico-chemical quality parameters during storage, but they showed a linear increase during the studied period. Conversely, the values of these parameters for the flavored samples (FOO) remained below their limits for peroxide value and K232, but they exceeded the limits for the free acidity and K270 parameters (Appendix A) [30]. The free acidity, which is mainly increased due to the hydrolysis of triacylglycerides to free fatty acids, was exceeded at the fifth month due to a higher water activity [31] and the migration of organic acids from herbs to the oil but not due to a higher oxidation. The limit for K270 parameters was exceeded by ROO samples from the third month onwards, which is not an anomalous result as it depends on the phenolic content in the oil and the bottling [12].

### 3.2. Virgin Olive Oil Shelf Life

The changes of the VOO samples throughout storage were first assessed, and later, the comparison with the FOO samples was carried out. More than 180 volatile compounds have been reported in VOO [38], and it is not only those with a higher concentration than their odor threshold that contribute to the aroma. In order to reduce the complexity of the system, only concentrations above the odor threshold were considered [38,41]. This study focused on several of the latter volatiles related to the aroma, and some of them, such as 3-pentanone or E-2-hexenal, are related to positive sensory perceptions. Others, such as E-2-heptenal or nonanal, are related to sensory defects. Table 1 shows the volatile compounds identified in the initial sample (VOO-0), their concentration (mg/Kg), odor threshold (mg/Kg) and associated sensory descriptors. The concentration of volatiles attributed to sensory defects in the initial sample were below their odor thresholds [35,42], therefore, they did not contribute to the sample aroma, which agrees with its classification as extra virgin olive oil.

The profiles of the volatiles changed throughout the storage, due to the modifications produced by the oxidation process. Initially VOO samples were rich in biogenically-formed C5 and C6 compounds—including 3-pentanone or E-2-hexenal—that can contribute to positive sensory perceptions [38]; the samples did not contain (or contained only at low concentrations) organic acids. After six months of storage, VOO samples presented a high number of compounds characterized by negative sensory descriptors such as acetic, propanoic and hexanoic acids, as well as other C7 and C9 compounds [38] such as heptanal or nonanal. The compounds generated throughout the storage were mainly due to the oxidation of unsaturated fatty acids. The study of the generation of sensory defects is crucial because when a VOO reaches high intensities of sensory defects, it is classified as lampante virgin olive oil and must undergo a refining process before it can be consumed [25], which contributes to economic losses.

All the studied compounds showed significant differences (*t*-test) (*p* < 0.05) during the storage, except for 1-butanol. Particularly, in VOO samples, the difference started becoming significant from the third month onwards (*t*-test). The behavior of each compound varied during the storage; compounds related to sensory defects increased while compounds related to positive perceptions decreased. Table 2 displays the information of several representative volatile compounds, of which the concentrations greatly differed during the storage, most of them being characterized as volatiles produced by the auto-oxidation of unsaturated fatty acids.

Pentanal is a volatile compound which is normally associated with a green fruity perception but it can also be related to a rancid off-flavor in olive oil undergoing oxidation [12]. Pentanal and hexanoic acid are good examples of a progressive increment during the study (Figure 1). Both compounds increased but with different slopes—higher for hexanoic acid, in part produced from hexanal oxidation, and lower for pentanal.

In addition to the compounds described in Table 2, there are other compounds (Table 1) that are also important because their sensory impact and their relationship with virgin olive oil quality as to be considered. Thus, 2-heptanone and 3-octanone, both related to sensory defects, increased their concentrations from 1.43 ± 0.01 to 11.01 ± 0.78 mg/Kg and from 0.04 ± 0.00 to 1.64 ± 0.12 mg/Kg, respectively. Other studied organic acids, such as acetic, propanoic and pentanoic acids, also increased between 5 and 30 times their initial concentrations. These results disagree with Cecchi et al. [11], who reported that the concentration of organic acids never exceeded 0.4 mg/Kg and the amount of acetic acid did not show a significant increase (*p* < 0.05).

The concentrations of all the volatile compounds were used to perform a cluster analysis, which allowed the classification of the samples according to the storage time. Figure 2 shows that the initial sample (VOO-0) and the samples from the first three months were clustered separately from the rest of samples, which formed other clusters where the most oxidized samples were included. The first cluster, however, was split into two groups: one group joins the initial and the first month samples, showing that both samples had fewer differences; and another group joins the samples corresponding to the second and third months, showing that they had differences in their volatile content due to the incipient progression of the oxidative process.

In the second cluster, the samples corresponding to the fourth month appeared separately from the other two more altered samples. The separation between the sample of the third month and the samples of the fourth, fifth and sixth months is in accordance with the information obtained from the *t*-test.

### 3.3. Comparison between Unflavored and Flavored Samples

The results of the study of the volatile compounds in VOO samples were compared with those obtained in FOO samples. The addition of rosemary and basil altered the content of some volatiles due to their antioxidant activity and the migration of volatile compounds. These changes varied not only depending on whether the sample was flavored or not, but also depending on the type of flavoring agent. The volatile composition of both vegetal materials was first identified to evaluate the potential migration of volatiles from the herbs to the oil. Table 3 shows the volatile compounds identified in the two vegetal materials: 23 in rosemary and 26 in basil. The presence of volatiles of the herbs in the FOO samples allowed the confirmation of the migration of several of them, but not all the basil and rosemary volatiles migrated from the herb to the oil; some volatiles remained in the herbs and others were already present in VOO, not all of which influenced the aroma.

Throughout the storage, several volatiles showed the same trend in FOO and VOO, such as pentanal, nonanal or ethyl acetate, but the rate of generation or degradation due to the oxidation varied between the different samples. As previously described, pentanal increased in VOO from the first month (38.25 ± 2.71 mg/Kg) to the sixth month (68.35 ± 4.83 mg/Kg), while in ROO, it increased from 33.90 ± 2.40 mg/Kg to 84.98 ± 6.00 mg/Kg, and in BOO it increased from 36.34 ± 2.57 mg/Kg to 70.86 ± 5.01 mg/Kg. The concentrations in the first months of storage were similar and increased to higher values; in the case of FOO samples, this was likely due to the migration of pentanal from the herbs to the oil and not only due to the auto-oxidation process.

The concentration of nonanal, in concordance with [43], increased during the storage even in flavored samples, but with different trend. Nonanal increased its concentration 10.91% in ROO and 34.93% in BOO, but these increments were lower than in VOO. Thus, nonanal—an aldehyde produced by the auto-oxidation of monounsaturated oleic acid—is a good example of the protective role of the herbs. While the concentration of nonanal increased significantly (*p* < 0.05) in VOO samples, it did not vary significantly (*p* < 0.05) in FOO samples; the concentration slope was higher in VOO than in FOO samples (Figure 3). In addition, the increase in the nonanal concentration was significant in VOO samples from the second month onwards, while the increase was delayed until the fourth month in FOO samples, especially in the case of ROO.

Other compounds that showed the same behavior in the three types of samples (VOO, ROO, BOO) were 2-heptanol and ethyl acetate. Thus, 2-heptanol showed an increasing trend, reaching the highest value in VOO in the sixth month. The concentration of ethyl acetate, however, decreased, with BOO samples having the highest amount (4.52 ± 0.32 mg/Kg); the difference between ROO and BOO could be due to a greater antioxidant protection of BOO.

On the other hand, compounds such as ethanol, 1-penten-3-ol (associated to positive sensory descriptors) and 1-octen-3-ol (responsible for the musty/mushroom sensory perception) [9] showed different behavior at least in one of the FOO samples. Ethanol showed a completely different behavior between FOO and VOO samples; as an average of FOO samples, its initial concentration was 19.17 ± 4.83 mg/Kg and increased to 24.32 ± 0.51 mg/Kg, but without significant differences (*p* < 0.05). 

Volatile 6-methyl-5-hepten-2-one showed a different behavior in each sample. In VOO, the concentration increased significantly (*p* < 0.05) due to the oxidation process [43], while the variations were not significant (*p* < 0.05) in FOO samples. In ROO samples, the concentration of 6-methyl-5-hepten-2-one showed a downward trend, whereas it did not change in BOO samples due to the decrease in its initial concentration which was compensated for by its transfer from the herb to the oil.

In relation to organic acids, the FOO samples showed large differences with VOO samples. Pentanoic and hexanoic acids maintained the same concentration as in the initial sample (VOO-0), indicating that there was a slowing down of the oxidative process that generated them due to the migration of antioxidant compounds. The concentration of propanoic and butanoic acids increased throughout the storage, but it remained below the concentration of the VOO samples. Additionally, a softer slope was observed (Figure 4), which implies that the generation of these acids was slower than in the unflavored VOO due to the protective role of the herbs against the oxidation process. Figure 4 also shows that ROO samples reached a higher concentration of butanoic acid than BOO samples, which supports the idea that basil has a greater protective role.

Multi-dimensional scaling was applied to all the samples using the volatiles previously described as variables (Figure 5). The results showed that the volatile compounds presented different concentrations depending on the storage time and the kind of flavoring. These differences were mainly due to the migration and effect of antioxidant compounds from the herbs to the virgin olive oil [37]. As mentioned above, it was from the third month onwards when significant differences began to appear, making it possible to separate older samples from the initial ones (Figure 5). The final two months of the ROO samples were also farther from the rest, but not as much as VOO-4 to VOO-6, while the BOO samples maintained similar concentrations, supporting once again the greater antioxidant capacity of basil compared to rosemary.

### 3.4. New Volatile Compounds

In addition to volatile compounds naturally existing in VOO and terpenes, rosemary and basil herbs have their own volatiles, for example, high concentrations of hydrocarbons and terpenes (Table 3). From the first month, FOO samples presented a different profile, justified by the transfer of these new compounds from the herbs to the oil. Some of the transferred compounds were present in both FOO (ROO, BOO), while others were unique of each one of them (Table 3). The presence of new volatiles in the oils is explained by their chemical properties and their capacity to move through biological membranes. The FOO samples did not show any decrease in these new volatiles throughout the study. On the contrary, most of the volatiles gradually increased their concentrations during storage due to the migration of the compounds from the herbs to the oil.

The common compounds identified in both herbs were γ-terpinene, limonene, 1,8-cineol and tetrahydrolinalool; both FOO samples showed an increase in their concentrations except for tetrahydrolinalool, the concentration of which remained unaltered during the storage. Depending on the flavoring, higher levels of terpenic compounds, such as limonene, were found [17]. The BOO samples showed significant (*p* < 0.05) differences between the first and sixth months of storage for γ-terpinene, which increased from 0.02 ± 0.001 mg/Kg to 0.04 ± 0.01 mg/Kg with significant differences (*p* < 0.05). Additionally, 1,8-cineol, which was not detected in the first month, increased to 0.04 ± 0.01 mg/Kg. The ROO samples, however, only showed significant (*p* < 0.05) differences for 1,8-cineol, the concentration of which increased from 0.01 ± 0.001 mg/Kg to 0.09 ± 0.02 mg/Kg. The increment in 1,8-cineol concentrations is important due to its impact on the sensory perception of the oil [44]. The concentrations of 1,8-cineol showed significant (*p* < 0.05) differences between the first and the sixth months in the ROO samples, but also between ROO and BOO samples.

In addition to the volatiles that are common to both herbs, there were others that were unique to each herb and migrated to the oil as well. The exclusive volatiles of rosemary herbs [45,46] that were quantified were m-mentene, α-terpene, β-caryophyllene, β-myrcene, camphene, borneol and α-pinene. The rosemary compounds whose concentrations were higher in ROO were m-mentene (6-fold increase) followed by α-terpinolene and camphene (5-fold increase). On the other hand, α-pinene, which has a fresh smell, could not be quantified until the fifth month, after a long time contact time between the herb and oil. The concentration of β-myrcene remained constant (0.22 ± 0.03 mg/Kg) throughout the storage. Table 4 reports that all the volatiles, except for β-myrcene, showed significant differences throughout storage due to the migration of the new compounds from the herb to the oil. The presence of all these new compounds is not only reflected in a very different volatile profile, but also in the aroma of the olive oil. Some of the compounds shown in Table 4 have been also described by other authors [38] while others, such as m-mentene, have not been previously described.

The volatiles that migrated from basil herbs to flavored olive oil (BOO) were different from flavored olive oil with rosemary herbs (ROO), except for the four compounds mentioned above. This resulted in different volatile and sensory profiles of the olive oils. Table 5 shows the main compounds that migrated to BOO samples from the basil herb. As in the ROO samples, none of the compounds decreased throughout the storage. However, β-farnesene was only detected in the samples from last two months, but in low concentrations, which may be due to the fact that this compound is mainly found in the root and in a lower concentration in the aerial part of the herb [47].

The other volatiles transferred from herbs increased their concentrations in BOO and ROO samples during the storage. It is remarkable that, except for 2-methylbutanal and dihydrolinalool, all the volatiles showed significant differences throughout the storage (Table 5).

### 3.5. Sensory Evaluation

The new volatile profiles influenced the sensory assessment of the resulting flavored oils (ROO, BOO). However, as already described, not all the volatile compounds transferred from the herbs to flavored oils contribute to the aroma. Thus, a sensory evaluation, using a descriptive–quantitative analysis, was carried to characterize the aroma of all the VOO and FOO samples.

Figure 6 shows the initial sensory profile of the EVOO sample (VOO-0). The sensory attributes green, olive fruit (green), tomato, apple and sweet quantitatively described the sensory profile of the non-stored VOO-0 sample as a typical profile of an EVOO without sensory defects.

Table 6 shows the qualitative sensory modifications suffered by the VOO and FOO throughout the storage; the quantitative data are shown in Appendix A. In the case of VOO, the initial profile was characterized by a gradual loss of the green and fruity nuances until the fourth month. At that time, the rancid sensory defect appeared for the first time and the intensity of the perception increased from then on, justified by higher concentrations of aldehydes and acids. The sensory profiles were different in FOO samples, though the presence of the herbs was detected from the first month. From the third month onwards, the sensory properties changed—decreasing the intensity of the perceptions in both herbs and becoming unpleasant in the case of BOO samples, which is in concordance with the physical—chemical analysis and the volatiles profiles. FOO samples also indicated that the presence of herbs in the oils from three to five months resulted in a masking effect of other perceptions. In the final month, assessors detected the aroma of dry herbs in the case of ROO and rancid began to be present in BOO, in accordance with the concentrations of volatiles in the flavored samples.

Considering the migration of volatiles from the herbs to the oil, the aroma of BOO samples was mainly due to β-pinene, ocimene and 1,8-cineol compounds. In the case of ROO samples, their aroma is due to the concentrations of camphene, β-myrcene, α-terpinolene, limonene and 1,8-cineol. Thus, the sensory assessment shows that the volatile compounds of the herbs modified the initial organoleptic characteristics of the virgin olive oil and mask other sensory perceptions, which needs to be further studied.

## 4. Conclusions

The application of DHS-GC in the study of the evolution of the volatile profiles of VOO and FOO for six months allowed the quantification of 27 volatile compounds that are responsible for their aroma. The volatiles were classified into two types: those responsible for the pleasant aroma, whose concentration decreased during storage; and others resulting from auto-oxidative or photo-oxidative processes that are responsible for the unpleasant aroma. The addition of rosemary and basil herbs to a VOO produced significant differences in volatile concentrations and a protective (or masking) effect was detected in rosemary and basil flavored oils. The volatile profile of rosemary and basil, identified by GC-MS, was mainly characterized by hydrocarbons and terpenes. Although some of them have already been quantified in the VOO, volatiles allowed distinguishing between flavored and non-flavored samples. Furthermore, the addition of basil or rosemary herbs to VOO enhances the initial VOO shelf life and helps to mask undesirable sensory perceptions. Further studies on the effect of rosemary and basil on virgin olive oil stored in darkness should be carried out and compared with the results obtained in this work, in addition to using different olive oil varieties.

The use of flavoring agents with a strong aroma that can mask the defects of VOOs can make fraud detection a difficult task, therefore, a future analytical challenge is the ability to guarantee an oil’s authenticity. Despite this problem, what remains to be addressed is the lack of legislation at the level of the European Union on the marketing of this type of flavored olive oil, which not only makes it difficult to market, but also creates helplessness in the protection and expectations for of consumers for this product.

## Figures and Tables

**Figure 1 antioxidants-12-00242-f001:**
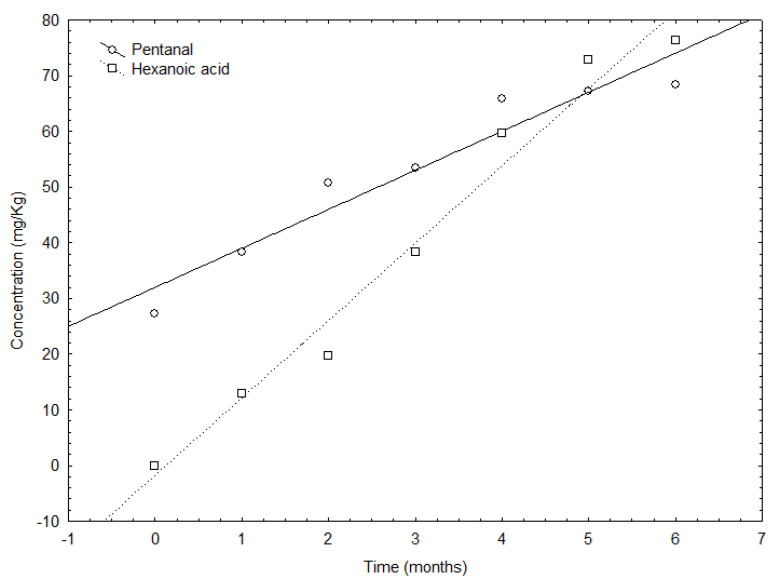
Evolution of pentanal and hexanoic acid concentration (mg/Kg) during the storage in VOO samples.

**Figure 2 antioxidants-12-00242-f002:**
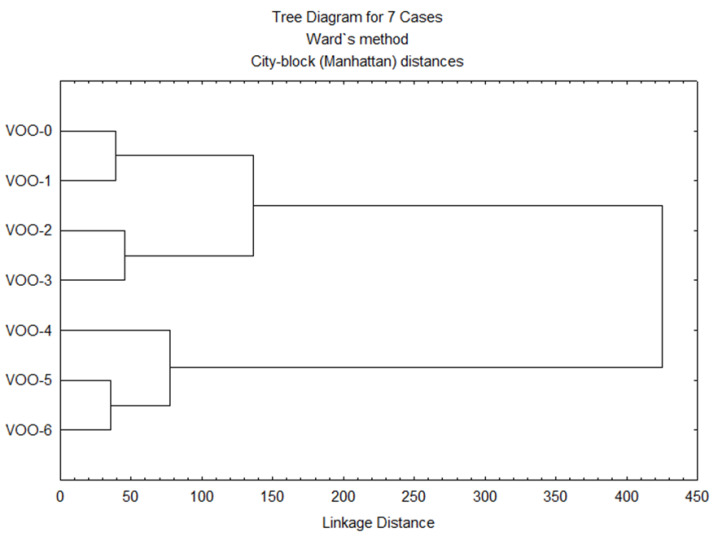
Cluster analysis of the volatile compounds of the VOO samples.

**Figure 3 antioxidants-12-00242-f003:**
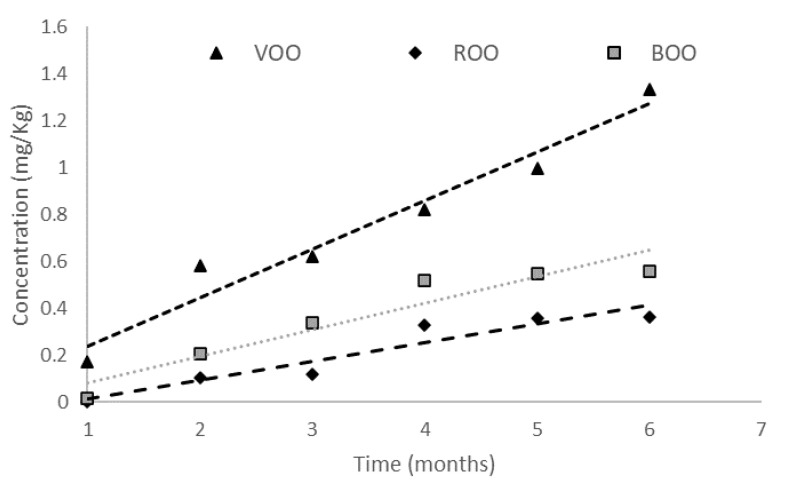
Variation in the concentration (mg/Kg) of nonanal in VOO, ROO and BOO samples throughout the storage.

**Figure 4 antioxidants-12-00242-f004:**
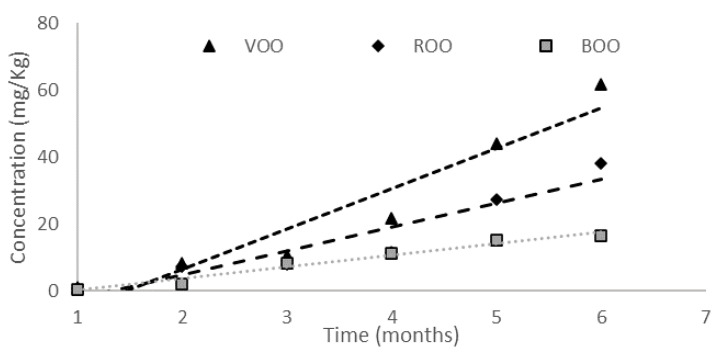
Concentration (mg/Kg) of butanoic acid during the storage in the three samples.

**Figure 5 antioxidants-12-00242-f005:**
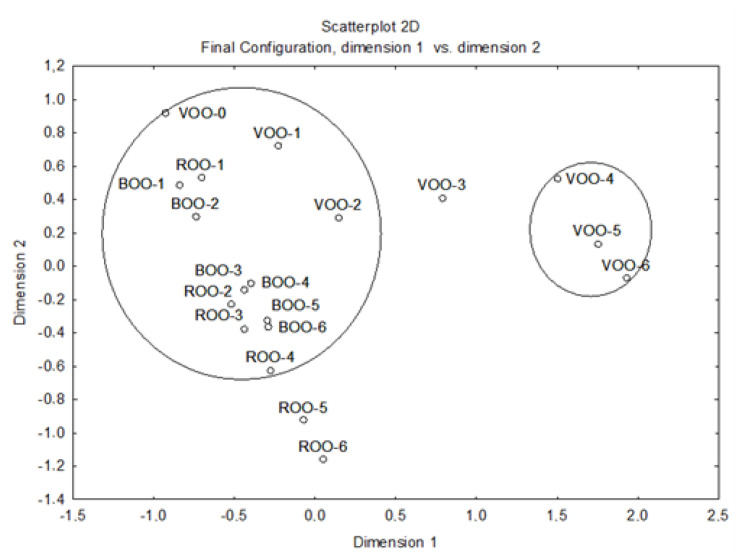
Multi-dimensional scaling (MDS) analysis applied to all the samples. VOO, virgin olive oil; ROO, virgin olive oil flavored with rosemary herbs; BOO, virgin olive oil flavored with basil herbs.

**Figure 6 antioxidants-12-00242-f006:**
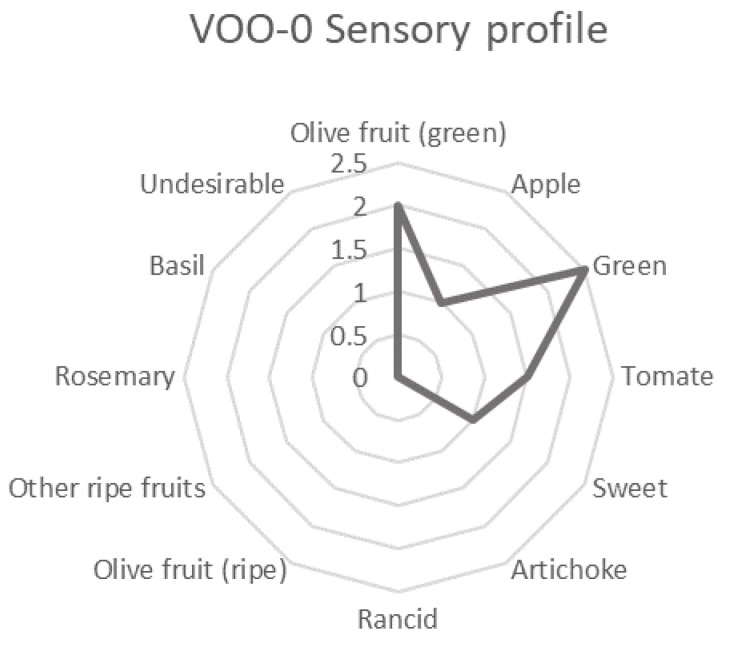
Spider chart of the sensory attributes evaluated in the non-stored extra virgin olive oil sample (VOO-0).

**Table 1 antioxidants-12-00242-t001:** Kovats index, retention time (min), concentration (mg/Kg), odor threshold (mg/Kg) and sensory descriptors of the volatile compounds of the sample VOO-0.

Volatile Compound	Kovats I	Rt ^a^ (min)	Conc. ^b^ (mg/Kg)	O. Thres. ^c^ (mg/Kg)	Sensory Descriptor [38]
Octane	800	4.126	0.71 ± 0.05	0.94	Alkane, solvent
Ethyl acetate	892	6.398	6.09 ± 0.43	0.94	Sweet, aromatic
Ethanol	932	7.457	29.51 ± 2.09	30.00	Apple, sweet, alcohol
3-Pentanone	964	8.271	0.36 ± 0.03	70.00	Sweet, fruity
Isobutyl acetate (Internal standard)	-	9.852	-	-	-
1-Penten-3-one	1016	11.548	0.16 ± 0.01	0.70	Spicy, mustard
Pentanal	969	12.873	27.30 ± 1.93	0.24	Oily, wood, bitter, almond
Hexanal	1074	14.479	0.87 ± 0.06	0.08	Green, green apple, grass, oily, fatty
Heptanal	1174	20.502	1.11 ± 0.08	-	Citrus, fatty
1-Butanol	1145	22.261	0.38 ± 0.08	0.40	Sweet, oily, drug
E-2-hexen-1-al	1216	23.440	2.59 ± 0.05	0.42	Fruity, green, bitter almond
2-Heptanone	1170	24.159	0.27 ± 0.10	0.30	Musty, soapy, cinnamon
2-Methyl-1-butanol	1210	24.949	2.50 ± 0.36	-	Ethereal, alcoholic, fatty
3-Octanone	1244	26.177	0.040 ± 0.001	-	Grass, musty, green, butter
E-2-heptenal	1282	27.552	4.14 ± 0.53	4.20	Soapy, fatty, almond, spicy
2-Heptanol	1288	28.299	0.01 ± 0.06	0.01	Mushroom, dirt, sweet
6-Methyl-5-hepten-2-one	1347	28.643	0.15 ± 0.01	1.00	Fruity, green, herb
E-3-Hexen-1-ol	1366	30.060	3.30 ± 0.23	1.00	Green grass
Nonanal	1385	31.467	Not detected	0.15	Rancid, citrus, soapy
1-Octen-3-ol	1390	33.115	0.54 ± 0.03	1.00	Musty, mushroom
Acetic acid	1452	35.110	0.48 ± 0.05	0.50	Bitter, winery, vinegar
Propionic acid	1527	38.316	0.61 ± 0.04	0.72	sweaty, pungent
Butanoic acid	1628	40.340	0.56 ± 0.09	0.65	Rancid, sweaty
Pentanoic acid	1720	44.162	0.33 ± 0.02	0.60	Spicy, rancid
Hexanoic acid	1829	49.000	Not detected	0.70	Rancid, cheese
Heptanoic acid	1990	53.450	Not detected	0.10	Rancid, fatty
Octanoic acid	2083	55.103	Not detected	3.00	Rancid, cheese, oily, fatty

Note: ^a^, Retention time; ^b^, Concentration; ^c^, Odor threshold.

**Table 2 antioxidants-12-00242-t002:** Concentration (mg/Kg; mean ± SD) of volatile compounds of the VOO samples during the six months of storage.

Compound	VOO-1	VOO-2	VOO-3	VOO-4	VOO-5	VOO-6
Pentanal	38.25 ± 2.71 ^a^	50.74 ± 3.59 ^b^	53.43 ± 3.78 ^b^	65.96 ± 4.66 ^c^	67.15 ± 4.75 ^c^	68.35 ± 4.83 ^c^
Nonanal	0.18 ± 0.04 ^a^	0.59 ± 0.09 ^b^	0.62 ± 0.16 ^b^	0.82 ± 0.17 ^b^	1.00 ± 0.21 ^c^	1.34 ± 0.26 ^c^
2-Heptanol	0.87 ± 0.06 ^a^	1.06 ± 0.08 ^a^	1.19 ± 0.08 ^b^	1.29 ± 0.09 ^b^	1.36 ± 0.9 ^b^	1.43 ± 0.10 ^b^
1-Octen-3-ol	0.52 ± 0.04 ^a^	0.52 ± 0.04 ^a^	0.53 ± 0.04 ^a^	0.55 ± 0.04 ^a^	0.61 ± 0.04 ^a^	0.80 ± 0.06 ^b^
Butanoic acid	1.14 ± 0.08 ^a^	8.26 ± 0.58 ^b^	10.52 ± 0.74 ^c^	21.80 ± 1.54 ^d^	43.87 ± 3.10 ^e^	61.66 ± 4.36 ^f^
Hexanoic acid	12.90 ±0.91 ^a^	19.74 ± 1.40 ^b^	38.24 ± 2.70 ^c^	59.57 ± 4.91 ^d^	72.94 ± 5.16 ^d^	76.31 ±5.40 ^d^
Ethyl acetate	4.96 ± 0.35 ^a^	3.36 ± 0.24 ^b^	3.14 ± 0.22 ^b^	2.82 ± 0.20 ^b^	2.35 ± 0.17 ^c^	2.13 ± 0.15 ^c^
Ethanol	26.38 ± 1.87 ^a^	24.44 ± 1.73 ^a^	23.76 ± 1.68 ^a^	20.81 ± 1.47 ^b^	20.83 ± 1.47 ^b^	20.85 ± 1.46 ^b^
1-penten-3-one	0.08 ± 0.01 ^a^	0.07 ± 0.01 ^a^	0.04 ± 0.01 ^b^	0.03 ± 0.00 ^c^	0.02 ± 0.00 ^d^	0.01 ± 0.00 ^e^

Note: Values with different letters (^a,b,c,d,e,f^) within the same row are significantly different (*p* < 0.05). Experiments were set up in two repetitions and samples for each repetition were analyzed in triplicate (*n* = 6).

**Table 3 antioxidants-12-00242-t003:** Volatile compounds identified in rosemary and basil herbs by SPME-GC-MS.

Rosemary Compounds								
Volatile Compound *	Kovats I **	Molecular Formula	Molecular Weight	Fragments (m/z)				
Octane	800	C_8_H_18_	114.23	43	85	57	71	56
Pentanal	935	C_5_H_10_O	86.13	44	29	58	41	27
α-pinene	1035	C_10_H_16_	136.23	93	92	79	43	94
Camphene	1080	C_10_H_16_	136.23	93	121	41	79	39
m-menthene	1150	C_10_H_18_	138.25	95	138	67		
p-menthene	1152	C_10_H_18_	138.25	139.1	97.1	95.1	113.1	83.1
β-myrcene	1158	C_10_H_16_	136.23	93	69	94	79	77
α-terpinene	1184	C_10_H_16_	136.23	121	93	136	91	77
Limonene	1208	C_10_H_16_	136.23	93	68	136	121	67
1,8-cineole	1222	C_10_H_18_O	154.25	43	81	108		
γ-terpinene	1262	C_10_H_16_	136.23	93	91	136	77	121
m-cymene	1267	C_10_H_14_	134.22	93	135	121	133	119
3-octanone	1285	C_8_H_16_O	128.21	43	57	29	72	99
α-terpinolene	1297	C_10_H_16_	136.23	93	121	136	91	79
3-octanol	1390	C_8_H_18_O	130.23	59	83	55	41	101
Tetrahydrolinalool	1397	C_10_H_22_O	158.28	73	69	55	43	129
Acetic acid	1452	C_2_H_4_O_2_	60.05	43	45	60	15	42
L-camphor	1498	C_10_H_16_O	152.23	152	108	81	95	109
Bornyl acetate	1610	C_12_H_20_O_2_	196.29	95	43	93		
β-caryophyllene	1618	C_15_H_24_	204.35	91.1	93.1	133.1	79.1	41.1
Borneol	1642	C_10_H_18_O	154.25	95	93	96	94	97
2,6-dimethyloctane	1727	C_10_H_22_	142.28	57	43	71		
Octadecane	1800	C_18_H_38_	254.50	71	85	99	70	113
**Basil compounds**								
**Volatile compound ***	**Kovats I ****	**Molecular formula**	**Molecular weight**	**Fragments (m/z)**				
2-methylbutanal	864	C_5_H_10_O	86.13	41	29	57		
3-methylbutanal	912	C_5_H_10_O	86.13	44	41	43	58	27
Pentanal	935	C_5_H_10_O	86.13	44	29	58	41	27
2,6-dimethyl-2-octene	936	C_10_H_20_	140.27	69	70	41		
Hexanal	1088	C_6_H_12_O	100.16	44	41	56	43	27
β-pinene	1118	C_10_H_16_	136.23	121.1	136.1	135.1		
Limonene	1208	C_10_H_16_	136.23	93	68	136	121	67
1,8-cineole	1222	C_10_H_18_O	154.25	43	81	108		
β-ocimene	1245	C_10_H_16_	136.23	93.1	91.1	79.1	92.1	77.1
γ-terpinene	1262	C_10_H_16_	136.23	93	91	136	77	121
p-cymene	1280	C_10_H_14_	134.22	119	134	91	120	117
6-methyl-5-hepten-2-one	1340	C_8_H_14_O	126.20	43	41	55	69	39
Tetrahydrolinalool	1397	C_10_H_22_O	158.28	73	69	55	43	129
Acetic acid	1452	C_2_H_4_O_2_	60.05	43	45	60	15	42
Copaene	1510	C_15_H_24_	204.35	105	91	41	119	93
Dihydrolinalool	1525	C_10_H_20_O	156.26	109	69	73	41	43
α-linalool	1544	C_10_H_18_O	154.25	71	93	55	69	80
β-farnesene	1674	C_15_H_24_	204.35	41	69	93	81	67
Aristolene	1697	C_15_H_24_	204.35	105	161	91		
Pentanoic acid	1698	C_5_H_10_O_2_	102.13	60	27	29	41	39
Naphthalene	1718	C_10_H_8_	128.17	128	127	51	64	126
2,6-dimethyloctane	1727	C_10_H_22_	142.28	57	43	71		
α-curcumene	1777	C_15_H_22_	202.33	119	132	41		
α-bergamotene	1779	C_15_H_24_	204.35	93	119	41	69	55
β-ionone	1955	C_13_H_20_O	192.30	177	43	41	178	135
Ethyl vanillin	2565	C_9_H_10_O_3_	166.17	137	166	138	81	27

* Identification of the rosemary and basil volatile compounds by mass spectrometry using the WILEY 7 (John Wiley & Sons Limited, NJ) library, and later confirmed by pure standards analysis. ** Kovats Index reported in literature [38].

**Table 4 antioxidants-12-00242-t004:** Concentration ± SD (mg/Kg) of volatile compounds from rosemary herb quantified in flavored virgin olive oil (ROO) during the six months of storage.

Time (Month)	M-Mentene	Α-Terpinolene	Β-Caryophyllene	Β-Myrcene	Camphene	Α-Pinene	Borneol
**1**	0.24 ± 0.01 ^a^	0.01 ± 0.00 ^a^	0.01 ± 0.00 ^a^	0.18 ± 0.00 ^a^	0.04 ± 0.00 ^a^	n.d. *	0.01 ± 0.00 ^a^
**2**	0.21 ± 0.04 ^a^	0.03 ± 0.00 ^b^	0.01 ± 0.00 ^a^	0.21 ± 0.02 ^a^	0.04 ± 0.00 ^a^	n.d.	0.02 ± 0.00 ^a^
**3**	0.84 ± 0.14 ^b^	0.03 ± 0.00 ^b^	0.01 ± 0.00 ^a^	0.21 ± 0.01 ^a^	0.14 ± 0.01 ^b^	n.d.	0.02 ± 0.00 ^a^
**4**	1.16 ± 0.08 ^b^	0.04 ± 0.01 ^b^	0.01 ± 0.00 ^a^	0.23 ± 0.02 ^a^	0.15 ± 0.01 ^c^	n.d.	0.02 ± 0.01 ^a^
**5**	1.38 ± 0.15 ^b^	0.05 ± 0.01 ^b^	0.02 ± 0.01 ^b^	0.24 ± 0.02 ^a^	0.20 ± 0.06 ^c^	0.20 ± 0.02 ^a^	0.02 ± 0.00 ^a^
**6**	1.43 ± 0.01 ^c^	0.05 ± 0.00 ^c^	0.02 ± 0.00 ^b^	0.25 ± 0.01 ^a^	0.32 ± 0.06 ^c^	0.27 ± 0.02 ^b^	0.02 ± 0.00 ^a^

Note: Values with different letters (^a,b,c^) within the same row are significantly different (*p* < 0.05). Experiments were set up in two repetitions and samples for each repetition were analyzed in triplicate (*n* = 6). * Not detected.

**Table 5 antioxidants-12-00242-t005:** Concentration ± SD (mg/Kg) of volatile compounds from basil herb quantified in flavored virgin olive oil (BOO) during the six months of storage.

Time (Month)	2-Methylbutanal	Naphthalene	P-Cymene	Β-Ocimene	Dihydrolinalool	Α-Linalool	Β-Farnesene	Β-Pinene
**1**	0.25 ± 0.07 ^a^	0.04 ± 0.01 ^a^	0.01 ± 0.00 ^a^	0.01 ± 0.00 ^a^	n.d.*	0.01 ± 0.00 ^a^	0.00 ± 0.00 ^a^	n.d.
**2**	0.33 ± 0.06 ^a^	0.07 ± 0.01 ^a^	0.02 ± 0.00 ^b^	0.01 ± 0.00 ^a^	0.01 ± 0.00 ^a^	0.01 ± 0.00 ^b^	0.00 ± 0.00 ^a^	0.01 ± 0.00 ^a^
**3**	0.40 ± 0.02 ^a^	0.08 ± 0.03 ^a^	0.02 ± 0.00 ^c^	0.02 ± 0.00 ^b^	0.01 ± 0.00 ^a^	0.02 ± 0.01 ^b^	0.00 ± 0.00 ^a^	0.03 ± 0.00 ^a^
**4**	0.40 ± 0.02 ^a^	0.11 ± 0.03 ^a^	0.02 ± 0.00 ^d^	0.04 ± 0.01 ^c^	0.02 ± 0.00 ^a^	0.02 ± 0.00 ^c^	0.00 ± 0.00 ^a^	0.04 ± 0.01 ^a^
**5**	0.42 ± 0.04 ^a^	0.10 ± 0.01 ^a^	0.04 ± 0.00 ^e^	0.04 ± 0.00 ^c^	0.02 ± 0.00 ^a^	0.02 ± 0.00 ^c^	0.01 ± 0.00 ^b^	0.14 ± 0.02 ^b^
**6**	0.43 ± 0.04 ^a^	0.13 ±0.02 ^b^	0.04 ± 0.01 ^e^	0.04 ± 0.01 ^c^	0.02 ± 0.00 ^a^	0.03 ± 0.01 ^d^	0.01 ± 0.00 ^b^	0.16 ± 0.03 ^b^

Note: Values with different letters (^a,b,c,d,e^) within the same row are significantly different (*p* < 0.05). Experiments were set up in two repetitions and samples for each repetition were analyzed in triplicate (*n* = 6). * Not detected.

**Table 6 antioxidants-12-00242-t006:** Qualitative sensory evaluation of the aroma of VOO and flavored (BOO, ROO) samples during the storage experiment.

Time (Month)	VOO	BOO	ROO
1	Fruity, green	Slightly basil	Rosemary
2	Slightly green	Basil	Intense rosemary
3	Slightly green, heated oil	Basil, unpleasant	Rosemary
4	Slightly green, slightly rancid	Slightly basil, unpleasant	Slightly rosemary
5	Slightly fatty, rancid	Slightly basil and altered	Slightly rosemary, dry herb
6	Pungent, rancid	Slightly basil, rancid	Rosemary, dry herb

## Data Availability

The data that support the findings of this study are contained within the article. More information is available from the corresponding author, M.T.M.

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
