# Peer review of "Effect of Flavorization on Virgin Olive Oil Oxidation and Volatile Profile"

_antioxidants, 2023, doi:10.3390/antiox12020242_

Round 1

Reviewer 1 Report

Revision of the manuscript “Effect of Flavorization on Virgin Olive Oil Oxidation and Volatile Profile”

The manuscript I reviewed aimed to produce olive oil flavoured with basil or rosemary and to compare the volatile profile of these oils with that of the non flavoured olive oil over 6 months of storage. A further objective was to evaluate the production of new volatile compounds in the flavoured oils and to correlate these new compounds with the samples’ shelf-life.

The manuscript is surely interesting, but several concerns arose to me during reading it. They are listed below and that have to be addressed in order to improve the quality of the manuscript before publication

Abstract:

The abstract is too much focused in describing the introductive part and the aim of the study (more than half of the abstract); then, the last part is focused on describing the conclusion of the work, while no results are reported. In my opinion, this abstract does not reflect the importance of the study, and it does not give to the reader any idea of the main results obtained. The abstract has to be strongly improved before the manuscript is acceptable for publication on Antioxidants

·         - Line 15: “Mild storage conditions”: what is the meaning of this sentence? Please, be more specific. I believe that storing the oil under light exposure for 12 h/die is a condition that favors the development of defects such as the rancid.

Introduction:

·         - Lines 39-51: in this part, the authors briefly summarized the main knowledge about the volatile compounds of virgin olive oil, citing a group of manuscripts dating back to the 90s of the last century. Some recent reviews have depicted the recent state of the art in the field of virgin olive oil volatile compounds, including both the information reported in the first manuscripts dating back to the 90s of the last century and the new findings reported in more recent manuscripts. Therefore, in addition to the manuscripts already cited in this part, at least either one recent review or some more recent manuscripts concerning this topic have to be cited.

In this context, I believe that the manuscript would benefit from an improvement of this part in which it emerges the great complexity of the volatile profile of virgin olive oil

·         - Lines 52-58: in this part, the authors briefly introduced the topic of the flavoured olive oil and the different types of flavouring processes. Two manuscripts have been recently published concerning these topics: one (https://www.sciencedirect.com/science/article/pii/S0924224422001352) is a review which gives a quite complete picture on the flavoured olive oils; the other one (https://www.sciencedirect.com/science/article/pii/S0308814622026589) is a study in which two of the flavouring methods cited by the authors (i.e., infusion of spices into the oil, joint malaxation of olive paste and herbs) are compared at mill scale. The authors must add these two manuscripts to the reference list, and a brief comment in this part of the introduction.

·         - Line 64: “Only a few of them”: which ones? Please add a few lines in the introduction (before the aims) in which the manuscripts focused on characterizing the evolution of flavored VOOs over time are described, so that it can be clear what is new in this manuscript with respect to the state of the art.

·         - Line 68: again, the authors used terms such as “storage under moderate conditions”. I suggest the authors to remove this type of wording and to give more specific indication of the condition of storage applied, here and throughout the manuscript

·         - Lines 64-70: please add in the aims of the study, which has been the infusion method used for producing the flavoured olive oils

Materials and methods:

·         - Lines 88: the regulation cited in the reference n° 25 (i.e., 2568/1991) is repealed. Please check and correct

·         - Lines 90-94: were the aromatic herbs infused into the olive oils for the whole 6-months storage period? Please specify

·         - The authors stated that they prepared six 100-ml bottles of non flavoured samples, six 100-ml bottles of samples flavoured with basil and six 100-ml bottles of samples flavoured with rosemary (lines 88-91), and that once per month a bottle from each group was samples over 6 months working in triplicate. Does this means that the authors performed triplicates of the following analysis with 100 mL of samples?

Ø  Free acidity

Ø  Peroxide value

Ø  Specific extinction values

Ø  Analysis of volatile compounds

Ø  Sensory analysis according to the IOC protocol with at least 8 panelists

Ø  Sensory evaluation by QDA with 8 assessors

·        - Line 109: please, add a figure with the evaluation sheet. It can be added either in the main manuscript or as a supporting information file.

·         - Lines 114-116: why the authors used two different techniques for the analysis of volatile compounds? Which was the analysis used for the quantitative analysis? Which one was used for obtaining the data reported in the results and discussion section?

·         - Lines 152-153. Which was the internal standard?

Results and discussion:

·         - Lines 177-188: please, add a table with the data of free acidity, peroxide value, K232 and K270 for the three types of samples over the 6 months of storage

·      - Lines 190-194: the situation is much more complex than so. The volatile compounds reported in VOO are much more than 180, as reported in a recent review. Furthermore, thanks to synergistic effects, not only the volatile compounds present at concentrations higher than their odor threshold contribute to the aroma. Please, improve this part

·         - Table 1: please indicate the name of the molecules used as internal standard

·         - Table 1: which are the references of the sensory descriptors reported?

·        - Table 1: is “Kovats I” the calculated Kovats index? If yes, please add a column with the Kovats index reported in the literature. Why the Kovats I for the internal standard was not reported?

·         - Table 1 and line 211: what is “1-heptenal”. Perhaps the authors meant heptanal?

·         - Line 236. Is it 3-octanone or 2-octanone?

·         - Lines 243-257 and Figure 2. I suggest to the authors removing this part, which is poor of information for the reader.

·         - Line 263: what is the meaning of “the type of flavorization”? Is it the type of flavoring agent (i.e., rosemary and basil)? Please, re-write in a more clear manner

·         - Table 3:

Ø  Please add a column with the retention index reported in the literature.

Ø  Is it necessary to report the fragments of the identified molecules?

Ø  I believe that a new column with quantitative data of the identified molecules would be very informative for the reader.

·         - Lines 274-277: these quantitative data for pentanal in virgin olive oil seems to be very high. Was the quantitative method validated? The authors should compare these values with those reported in the literature

·        - Lines 338-340: terpenes are also naturally present in virgin olive oil, as reported mainly by the works of the group of Stefania Vichi of the University of Barcelona. Their presence mainly depend on the olive variety, and it is likely that the variety for the oil used for this study does not contain significant amounts of terpenes. Please, re-phrase this sentence in this sense.

·         - Table 5: 2-methylbutanal is a typical molecule present in virgin olive oil, mainly when it is defective for defects from microbiological origin. Are authors sure that this molecule was transferred into the oil from the basil? Please comment.

·         - Line 399: what is the meaning of the attribute “sweet” for an olive oil sample? It is strange to me to think to the sweetness during tasting an olive oil

·         - Line 413: the authors performed only chemical analysis and not the physical ones. Please correct, here and throughout the manuscript

·         - Lines 405-417: It is well-known that the storage of the oil under light exposure accelerates the development of sensory defects such as the rancid and the increase of some volatile compounds linked to this type of defect. To investigate on the differences in storing these type of flavoured oils under light exposure or in the dark could be the matter of a future research integrating that described in this article, and the authors must add a few lines to discuss about this aspect in this part of the manuscript (or otherwise, in the conclusion section).

Conclusion:

This section seems more a short summary of the manuscript than a real conclusion section. I believe authors should improve this part discussing which are the main conclusion gained thanks to their work, which are the possible future studies to make a step forward.

Reviewer 2 Report

The article presents some interesting information about aroma profilile of VOO flavoured by two herbs during storage. Some additional information as well as data should be add to present form of manuscript.

- please clarify the design of experiment: in methods there is information that each type of oil was divided in 6 samples which were bottled and stored for 6 months and analyses were done after each month. that means that there was no repetitions done and only repetitions were connected with number of analytical measurements of sample from the same bottle; on the other hand information under tables is that experiments were set up in two repetitions?

- please explain why such storage conditions were used

- results of quality chemical parameters should be presented in table as well

- please add table showing changes in main volatile compounds of VOO

- please add data of "0" sample (fresh sample - pure as well as with herbs extracts)

- please add spider charts of flavoured oils

Reviewer 3 Report

Comment

Many notable papers have been published regarding the addition of herbs to olive oil that the authors did not consider as: (Issaoui et al. (2016), Natural Product Communications, 11(6), 775-780; Baiano et al. (2005), JAOCS, 82(10), 759-765; Boulares et al. (2022), Rivista Italiana delle Sostanze Grasse, 99(2), 131-140 etc).

1.       Lines 68, 89, 95. Not 6 but six

2.       Line 92. Please provide geographic coordinates and harvest year.

3.       Line 136. Please provide the names of pure standards.

4.       Figures 3,4. Please add the trend lines for VOO, BOO, ROO

5.       Line 304. Because of the standard deviation of the original concentration, I find the increase unclear. The same observation applies to γ-terpinene (lines 356-357).

6.       Tables 1,4,5. Select not detected or n.d. for the tables

Round 2

Reviewer 1 Report

The authors significantly improved the manuscript according to the suggestions of the reviewers.

Reviewer 2 Report

I have no additional comments to provided version of article